# Estimating the Fiscal Consequences of National Immunization Programs Using a “Government Perspective” Public Economic Framework

**DOI:** 10.3390/vaccines8030495

**Published:** 2020-09-02

**Authors:** Mark P. Connolly, Nikolaos Kotsopoulos

**Affiliations:** 1Global Market Access Solutions, Health Economics, St-Prex, 1162 St-Prex, Switzerland; nikos@gmasoln.com; 2Unit of PharmacoEpidemiology & PharmacoEconomics, Department of Pharmacy, University of Groningen, 9713 Groningen, The Netherlands; 3Department of Economics, University of Athens, 105 59 Athens, Greece

**Keywords:** public economics, immunization, economic evaluation, vaccination, cost-benefit analysis, fiscal, lifetime, health shock

## Abstract

Infectious diseases can impose considerable mortality and morbidity for children and adult populations resulting in both short- and long-term fiscal costs for government. Traditionally, healthcare costs are the dominant consideration in economic evaluations of vaccines, which likely ignores many costs that fall on governments in relation to vaccine-preventable conditions. In recent years, fiscal health modeling has been proposed as a complementary approach to cost-effectiveness analysis for considering the broader consequences for governments attributed to vaccines. Fiscal modeling evaluates public health investments attributed to treatments or preventive interventions in the case of vaccination, and how these investments influence government public accounts. This involves translating morbidity and mortality outcomes that can lead to disability, associated costs, early retirement due to poor health, and death, which can result in lost tax revenue for government attributed to reduced lifetime productivity. To assess fiscal consequences of public health programs, discounted cash flow analysis can be used to translate how changes in morbidity and mortality influence transfer payments and changes in lifetime taxes paid based on initial health program investments. The aim of this review is to describe the fiscal health modeling framework in the context of vaccines and demonstrate key features of this approach and the role that public economic assessment of vaccines can make in understanding the broader economic consequences of investing in vaccination programs. In this review, we describe the theoretical foundations for fiscal modeling, the aims of fiscal model, the analytical outputs, and discuss the relevance of this framework for evaluating the economics of vaccines.

## 1. Background

Health shocks, such as the onset of chronic diseases and chronic health conditions, can have substantial fiscal consequences for governments in relation to future disability payments as people discontinue work and subsequently pay fewer taxes [1,2]. In developed economies, social support mechanisms can help to maintain living standards for people experiencing a catastrophic health event including those attributable to infectious disease. Additionally, health conditions in developed economies can often lead to lower lifetime wealth accumulation portending dependence on government assistance [3]. In low- and middle-income countries where fewer social support programs exist, the costs of health events are absorbed by families, often leading to catastrophic expenditure which can have a detrimental impact on a household’s future economic prospects [4]. Events associated with vaccine-preventable infectious diseases can be deemed as severe health shocks with multiple economic effects for societies, governments, healthcare systems, households, and individuals that should be considered.

From a public economic or fiscal perspective, every person of working age that is not working and is receiving public benefits relies on the output of remaining workers. The necessity for raising additional taxes to pay for those afflicted by illness and unable to work raises inefficiencies as it imposes consumers’ and producers’ surplus losses for the economy. The same can be said for premature mortality in working-aged adults as this removes individuals from the pool of available workers paying taxes. As taxes are raised to pay for social programs, this alters prices and quantities of goods and services demanded in the economy with further lower utility for consumers with resulting fewer products sold for manufacturers. Some estimates suggest that the deadweight loss attributable to ill-health in the economy can reach 28% of each tax dollar collected [5]. Considering the inefficiencies attributed to changing health conditions and the resulting deadweight loss provides a strong motivation for understanding how investments in healthcare can not only prevent deadweight losses, but also how these investments influence sustainability through estimation of short- and long-term costs for government and taxes gained from these investments.

Fiscal consequences can be attributed to a range of vaccine-preventable infectious diseases in children and adults [6,7]. The specific consequences are age-dependent, however, where even premature death can have future fiscal consequences as these individuals do not achieve their full economic lifecycle potential and governments are denied the opportunity to tax their future earnings [8]. In contrast, many savings can occur for governments from premature mortality due to avoiding future transfer payments and pension costs. Considering the nature of individual financial transactions between individuals and the state, it is important to explore how vaccination can give rise to different fiscal conditions through preventing future infectious diseases.

Government investments in national immunization programs represent national infrastructure investments that contribute to building and maintaining a healthy work force and avoiding short- and long-term health consequences. Through this lens, we can explore investments in national immunization programs using established economic appraisal methods for assessing the short- and long-term fiscal consequences for government attributed to immunization programs. Whilst the analytic approach shares many methodological similarities with conventional health economic analyses, there are several deviations and analysts should see this approach from a unique perspective, namely, to explore the cross-sectorial impact across different government sectors following investments in national immunization.

Conventional health economic frameworks applied to vaccines adopt a utilitarian perspective to conduct incremental cost-effectiveness analysis between alternatives. Being incremental, these analyses only consider costs and utility benefits that are directly related to the alternatives under study. Governmental costs and benefits are deemed as transfers and are hence excluded. The deadweight loss that can result from governmental transfer payments attributed to the disease is not considered. Historically, incremental cost-effectiveness analyses have proven to be valuable tools for allocating resources across several new, mostly therapeutic interventions, offering incremental innovations whilst competing for constrained budgets.

Unlike therapeutic interventions, vaccines can be deemed as a radical innovation offering benefits that will materialize years after vaccination. More importantly, vaccination programs can generate positive externalities due to the reduction of disease transmission which may, in turn, result in multiplicative broader economic effects which are not captured by cost-utility analyses [9]. However, the funding of national immunization policies is mostly tax-financed and operates with fiscal constraints [10,11]; hence, they are governed by different financial principles than those of conventional health economics. In no area of medicine is this more acutely observed than with national vaccine programs in which members of the Ministry of Finance are often involved with decision making due to the large acquisition costs. The fiscal modeling approach addresses the needs of financially focused stakeholders, helping them to understand how preventing future infectious diseases can influence government transfer and tax receipts. Fiscal elements can also be incorporated into optimization models, applying fixed budget constraints to help prioritize the introduction of vaccines [12].

In 2018, the International Society for Pharmacoeconomics and Outcomes Research (ISPOR) Good Practices for Outcomes Research Task Force Report outlined best practice guidelines for conducting fiscal health modeling in vaccines [13]. In this paper we describe the theoretical background to fiscal modeling, the benefits of this approach in establishing the value of vaccines, and outline how to employ fiscal models to inform resource allocation decisions.

## 2. Fiscal Model Framework

Recent investigations have highlighted how poor health can influence governments, both in terms of increased spending on transfer programs from government to citizens, i.e., disability costs, allowances, and social insurance costs for increased absenteeism, and lost tax revenues attributed to premature mortality [14,15,16]. The fiscal modeling framework applied to health and healthcare interventions involves understanding how changes in health status can influence the life trajectory of individuals and/or cohorts and how these variances influence public accounts compared to population averages; for instance, events such as premature mortality and disability which influence a person’s ability to work and pay taxes, while increasing their dependence on public benefits programs [1]. The first stage involves modeling the typical or average fiscal life course using established public economic frameworks such as generational accounting which account for age-specific per capita taxes and likelihood for receiving public benefits [8,17]. Within this public economic framework, it is possible to evaluate national immunization policies and vaccine investment choices and the resulting rates of infectious disease and resulting morbidity and mortality. The output is then applied to the fiscal framework to assess the impact that communicable diseases can have on productive capacity, consequently influencing future tax revenue for governments and transfer payments [2,18]. This also includes mortality in children which represent the future workforce and taxpayers.

The fiscal modeling framework treats public health costs associated with a vaccination program as investments in population health that can give rise to future fiscal costs and benefits for government. Similar to other economic evaluations, the main analytic tools originate from finance and include the combination of discounted financial flows to produce indices such as the return on investment (ROI), the net present value (NPV), and/or the benefit cost ratio (BCR) [19]. In economics, and for society in general, it has been established that people have a positive time preference for benefits sooner rather than benefits that are likely to occur in the future; therefore, it is necessary to reflect future costs and benefits in today’s value by discounting [20]. Discounting future costs and benefits enables policymakers to make decisions today by valuing all future costs and benefits based on values in present terms. In the analysis described here, changes in future taxes and transfer payment that occur every year attributed to a vaccine investment are discounted. The discount rate selected is an important parameter as costs of vaccination are up-front, but the economic benefits occur in the future [21]. The choice of discount rate varies by country and health economic guidelines, but generally ranges from 3% to 5% [22]. Future fiscal costs can be associated with government transfers such as disability payments, living allowances, and pensions, whilst fiscal benefits include future direct and indirect taxes and statutory social insurance payments linked to earnings paid by citizens.

Vaccination costs represent the public health investment undertaken by the government, typically during the first year of the analytic time horizon, i.e., year *t*_0_. For governments, fiscal costs and benefits represent annual cash flows that can be converted into present values using discounting and subsequently aggregated to calculate the NPV of the fiscal effect of a specific vaccine. In the equation below, the duration of the analysis post-vaccination is noted by (*T*). The time horizon (*T*) to apply in the analysis is dependent on the relevant research question but should be long enough to capture the full range of benefits linked to vaccination. Over the analytic time horizon, the sum of all taxes paid (*R_t_*) over the analytic time horizon (*T*) is captured. In the fiscal analysis, tax revenues (*R_t_*) are deducted by the amount of transfers received (*E_t_*) to establish the net fiscal effect of changes in morbidity and mortality (*R_t_* − *E_t_*). Furthermore, the costs of the initial vaccine investment are deducted from net taxes as this represents public investment for government (*K_o_*). Investing in vaccines leads to changes in population morbidity and mortality, and these changes influence the amount of tax collected by government from the population and the amount of transfer payments; hence, deducting the vaccine costs (*K_o_*) demonstrates for government the trade-offs associated with vaccine costs, and determines the government return on investment that a specific vaccination yields (Equation (1)). In addition to net tax, a breakeven year post investment in vaccination can be calculated which highlights the year in which all investment costs (*K_o_*) have been paid for through future taxes and/or reduced transfer payments in relation to the investment. Simply put, the breakeven age is when *R_t_* − *E_t_* − *K_o_* = 0. Equation (1):(1)$$Net\;tax=\sum_{t=0}^{T}\left(\frac{R_{t}{-E}_{t}}{{\left(1+r\right)}^{t}}\right)-K_{o}$$
*R_t_*, sum gross tax revenues accruing from the individual in year *t*; *E_t_*, sum of government transfers on the individual in year *t*; *r*, discount rate used to convert flows into present values; discounting formula (1 + *r*)^−*t*^. The discount rate is applied to all costs that occur in each year over the analysis timeframe. *T*, time horizon of the analysis, e.g., average life span; *K_o_*, vaccine investment costs in year *t*_0_.

In the absence of data for governmental transfers, a fiscal analysis based on gross tax can be conducted which converts lifetime productivity, which health economists typically measure in terms of total lifetime earnings, to lifetime tax revenue. Return on investment from a specific vaccine can also be estimated using incremental analysis that compares the fiscal effects of homogeneous vaccinated and unvaccinated cohorts. In incremental analyses, the net tax illustrated in Equation (1) can be estimated for vaccinated and unvaccinated cohorts with the differential representing the fiscal benefit of a vaccination. The results of an incremental analysis can also be represented as a ratio of the fiscal benefits to vaccination costs, i.e., as a fiscal benefit cost ratio (fBCR). Fiscal analyses require similar epidemiological and clinical evidence as the evidence used in cost-effectiveness evaluations of vaccines. In addition to health costs, fiscal analyses require country-specific, annualized macroeconomic evidence for tax revenues, transfer costs, and often data on potential broader, cross-sectorial economic consequences. Although much of the data needed may not be readily available in the published literature, there are countries that report in detail diachronic data for governmental perspective revenues and costs [13]. Moreover, household surveys can be a useful source of primary data that can be analyzed to inform fiscal analyses.

To illustrate the application of the equations, let us assume that a cohort of unvaccinated children has a lifetime present value of projected tax revenues of $1 million and public transfers of $0.5 million. We now introduce a vaccination that costs $0.2 million. The changes in morbidity and mortality attributed to the vaccine for the cohort gives rise to projected present value of lifetime tax revenues for the vaccinated cohort of $2 million and the corresponding transfers would be $0.3 million. The net tax for the unvaccinated cohort would be $1 million in taxes minus $0.5 million in transfers, generating net taxes of $0.5 million. By comparison, the net tax for the vaccinated cohort would be $2 million in taxes minus $0.3 million minus $0.2 million in vaccination costs, giving rise to $1.5 million in net tax. The incremental fiscal effect of vaccination would be the net tax of vaccinated minus the net tax unvaccinated cohorts: i.e., $1.5 million (vaccinated) minus $0.5 million (unvaccinated) = $1 million.

Previous studies have explored the fiscal impact of national strategies in children and adults for a variety of infectious conditions. Whilst vaccination was used to prevent different conditions, there are emerging trends that can help to inform how vaccines influence public economics. In the following section we discuss several prior applications of fiscal modeling applied to vaccines.

## 3. Rotavirus Vaccination in Children

Rotavirus is a highlight contagious virus causing seasonal diarrhea mostly in children younger than 1 year of age and is responsible for more than 200,000 annual deaths mostly in developing countries [23]. As the condition mostly strikes younger children, the related healthcare costs mostly occur in early years of life. The fiscal consequences of rotavirus will vary depending on the setting in which the cases occur, whereby higher mortality is often observed in developing countries, in comparison with developed countries where high healthcare resource costs can be incurred but with lower mortality.

The fiscal consequences of rotavirus have been assessed in previous investigations in low- and middle-income countries where interested readers can explore the model inputs in more detail [24,25]. The fiscal impact of introducing rotavirus vaccination in Egypt was estimated using a cohort model simulating the lifetime fiscal costs and benefits of vaccinated and unvaccinated birth cohorts. The modeling study showed that the introduction of rotavirus vaccination generated considerable immediate health cost-savings. Over longer time horizons, the dominant fiscal benefit was attributed to reduced mortality which influenced the size of the vaccinated cohort, resulting in increasing numbers of future taxpayers. The analysis conducted for Egypt indicated modest healthcare cost savings and increased fiscal revenue, achieving a breakeven age of 22 years when the net effect associated with the initial rotavirus vaccination costs became neutral, i.e., fBCR = 0.0 [25]. Over time, the fBCR increased, achieving 1.2 as the cohort of Egyptian children reached aged 25, and 2.8 when the cohort reached the age of 50. Subsequent studies applying similarly designed rotavirus fiscal models were developed for Ghana and Vietnam, showing positive net effects with the breakeven ages reported being ages 33 and 43, respectively [24]. Cross-country differences in breakeven ages reflect local demographic and socioeconomic conditions. Reported differences in achieving positive net effects in different countries reflect the different economic conditions across countries in which vaccine prices vary, differences in the age of starting work, wages, and earnings growth, tax burden differences, and high inflation rates which can influence discount rates applied in the models. This highlights the importance of considering each vaccine investment in relation to the unique economic conditions to understand the likely fiscal consequences associated with the investment.

A similar, however, broader analytic framework was used to estimate the consequences of rotavirus vaccination in a developed economy. A macroeconomic analytic framework namely, social accounting matrix (SAM) was applied to the Netherlands. The SAM framework encapsulates fiscal effects; however, it simultaneously explores a wider variety of economic perspectives and the interaction of these sectors [26]. The economic principle which the framework was based on is that in developed economies; parents often forego work to care for sick children which consequently reduces their earnings while caring for a sick child [27]. Within the SAM framework, lost earnings for parents translates into reduced income taxes and lower disposable income which reduces future spending and consequently collection of fewer taxes by government [27]. The SAM framework vaccination program provides an approach that could have more widespread applicability in vaccines as it reflects multiple perspectives within a single framework [26].

## 4. Human Papillomaviruses

Human papillomavirus (HPV) is a highly contagious condition with many serotypes which are spread by skin-to-skin contact. The disease can manifest differently depending on the serotype and the site of infection which can range from genital warts in males and females to vaginal and cervical cancer in females and penile cancer in males [28]. The likelihood for HPV-attributable cancer can give rise to numerous fiscal consequences in males and females due to increased morbidity and the potential of early mortality [29].

The fiscal consequences of introducing HPV immunization as part of the universal vaccination program in 12-year-old males and females was explored in Germany, based on the likelihood for preventing future head and neck cancers. A cohort model which had previously been used to estimate the cost-effectiveness of HPV vaccination was employed to simulate the long-term impact of HPV vaccination on future morbidity and mortality. Based on cohort simulations, it was projected that vaccination could prevent 857 HPV cancer deaths and 1527 cervical cancer cases and 630 other cancers in addition to 45,000 fewer cases of genital warts. These morbidity and mortality benefits were translated into cost-savings for both the healthcare and the social insurance system as well as increased productivity and therefore tax revenue gains for the government. Investing in vaccination against HPV was estimated to generate positive fiscal effects. Specifically, the resulting fBCR for the vaccinated cohort suggested that investing €1 in universal HPV vaccination in Germany could yield €1.7 in tax revenue gains over the lifetime of the cohorts. The fiscal return from girls (€3.1 for every €1 invested) was significantly higher than in boys (€0.3 for every €1 invested) which reflects the higher morbidity and mortality in females. In this respect, the increased fiscal benefits from girls appear to subsidize investment costs in boys which had a lower fiscal yield, yet with a fiscal gain.

## 5. Adult Immunization

Preventing disease burden in the working-age adult population may result in substantial macroeconomic benefits that can be directly linked to economic growth [30]. Discounted economic analyses favor short-term over long-term benefits; thus, the benefits of adult vaccinations which materialize within a shorter time horizon compared to pediatric vaccinations may correspond to higher discounted effects. Vaccinating adults can give rise to different fiscal consequences compared with children based on timing in relation to the fiscal life cycle that depends on the age at which vaccination occurs and the likely age for preventing infectious diseases. Preventing infectious diseases in working-aged adults will reduce short-term absences from work and prevent early mortality and in some cases reduce or even prevent disability benefits. For governments, premature mortality results in reduced lifetime taxes collected from these individuals; however, this can also lead to savings for government from avoiding future pension payments.

A study on behalf of the industry body Vaccines Europe and the Supporting Active Ageing through Immunisation (SAATI) partnership was reported in 2012. This study explored the fiscal consequences associated with a portfolio of adult vaccinations in the Netherlands. Applying a government perspective, this study investigated how adopting a life-course approach to immunization as part of healthy ageing policies and implementing immunization programs for adults aged above 50 can influence the clinical and economic burden of infectious diseases. It was reported that implementing comprehensive adult immunization programs for seasonal influenza, diphtheria, tetanus, pneumococcal diseases, pertussis, and herpes zoster was estimated to cost €136 million which included costs for annual influenza [31]. The comprehensive immunization schedule in adults was projected to prevent approximately 34,500 infectious disease cases and prevent an estimated 5700 deaths. Fiscal projections indicated that every €1 spent on adult immunization yielded €4 of benefit for government attributed to medical cost-savings (€6.6 million), reduced social insurance payments (€4.2 million), and reduced disability costs (€0.5 million) [31]. Furthermore, labor productivity gains in the adult cohort would result in future tax gains for government (€537 million) [31]. The fiscal model framework helps to illustrate the additional cost elements observed when evaluating health programs beyond simply health costs.

## 6. Discussion

To put into context the relevance of fiscal analysis in healthcare, it is important to understand the magnitude that ill-health can have on government public accounts. In 2008, a report commissioned by the UK government described the impact of ill-health in working-age adults where it was reported that healthcare costs represented only 8–15% of total government costs in individuals with poor health. It was reported that the main cost driver for government, attributed to ill-health, was associated with social benefit programs, i.e., transfer costs and lost tax revenue, reaching approximately 85–90% public costs [14]. Furthermore, with regard to the economics of vaccines, it has been argued that neither cost-effectiveness nor cost-benefit analysis takes full account of the broader economic impacts of immunization [32]; hence, there is a need to consider alternative frameworks for capturing vaccine value. The fiscal modeling framework seems well equipped to help understand the broader consequences of vaccination costs and other health programs and their impact on government revenues.

The fiscal modeling framework applied to health interventions is based on the generational accounting methodology, which starts from the premise that all government spending must be paid for by current or future generations [17]. This fiscal reality underscores issues of generational fairness and whether each generation can pay for various government benefits received over their lifetime and whether some generations may receive more public benefit than they have paid for over their lifetime. Within the fiscal modeling framework, we can partially answer this question by investigating whether introducing a vaccine to a cohort of individuals will in fact be paid for by the cohort through future tax contributions. In this respect, it is less important to look incrementally at program costs, but rather explore endogenously whether the generation/cohort vaccinated or treated will pay for the interventions through future tax contributions, or in the case of older cohorts from their residual lifetime tax contributions. Furthermore, since we explore the fiscal return on capital attributed to government program spending, we can explore issues of sustainability as we consider future spending and revenue implications simultaneously in relation to allocation decisions. If generations are not able to pay for their own interventions through lifetime tax contributions, then it is reasonable to ask whether it is fair that future generations should be expected to pay for these healthcare benefits. When the fiscal modeling approach is applied to vaccines, it is easy to explore this notion because all members of the cohort are theoretically at risk and eligible for vaccination and all have the capacity to benefit from vaccine investments.

The fiscal modeling framework informs our understanding of how investments in vaccines influence public accounts which can be used to set healthcare priorities. The findings from fiscal models enrich our understanding of conclusions drawn from cost-effectiveness studies and sometimes reach different conclusions compared to cost-effectiveness analysis as previously reported [33]. Nothing is lost by employing fiscal models for evaluating vaccines, and everything is to be gained. Firstly, it helps those paying for national immunization programs to understand how public accounts are influenced by vaccine investments. Costs can then be considered in relation to revenue, i.e., taxes. Secondly, the timeframe over which benefits accrue over generations are neatly laid out as are how changes in health outcomes influence government public accounts. Thirdly, in countries operating with fixed budgets, constrained optimization modeling can be combined with fiscal models to inform allocation decisions that maximize fiscal returns [12,13]. Governments can then prioritize vaccines based on the likely funding stream and timing that results from the allocation of resources. This is likely an important consideration for countries that are graduating from reliance on Global Alliance for Vaccines and Immunisation (GAVI) funding for vaccines in which co-financing and tax financing will be key components in building sustainable funding mechanisms for vaccines [10,34].

The basis for conducting economic analyses of healthcare interventions is to optimize outcomes with limited resources. In the era of cost-effectiveness analysis, optimizing efficiency requires setting cost-effectiveness thresholds or using league tables to allocate resources whilst independently assessing budget impact. In practice, few agencies universally apply cost-per-outcome thresholds to allocate resources, and for those agencies with approximate benchmarks, there can often be variation in accepted thresholds. This would suggest that additional information can enrich and inform allocation decisions. To some extent, the fiscal modeling framework applied to medical technologies provides an operational decision-making metric, whilst simultaneously assessing the budget impact of new interventions. Importantly, fiscal modeling considers cost more broadly than traditional budget impact models by taking the cross-sectorial government budget impact based on changes in morbidity and mortality into consideration. These costs are then simultaneously combined in relation to capital investments, i.e., vaccine program costs, to measure the impact for government, and the value is captured using the benefit–cost ratio based on the investment capital. Although there are no established criteria for allocating resources, higher benefit-cost ratios are better than lower. Furthermore, as vaccines represent governmental investments, the results can be compared with other public expenditure programs. We advocate that fiscal models have a place for informing allocation decisions in healthcare which are complementary to current economic frameworks used for allocating resources.

The most informative output from fiscal analysis is the fBCR which simultaneously captures incremental gross taxes and incremental public benefits in relation to vaccine investment cost. Immunization can change morbidity and mortality which can translate into productivity gains and, hence, lifetime taxes paid. From this, the public benefits, i.e., transfers, are deducted from incremental gross tax gains to obtain the net fiscal position which is divided by the vaccination program costs. This metric enables decision-makers the opportunity to understand how each monetary unit invested translates into future net tax revenue for government. Each fBCR is specific to the country for which the analysis was performed and reflects the disease prevalence, vaccine efficacy, wages, tax system, and public benefits for each country; therefore, fBCR ratios are not readily transferable to other countries.

The examples described here illustrate the fiscal benefits of established vaccination programs for adults and children. This work raises questions about what can be said fiscally regarding infectious diseases for which no established vaccine is available. Clearly, the attributable morbidity and mortality from these conditions will cause untold fiscal consequences that influence governments. When infectious diseases are viewed from the fiscal framework, it is easy to justify why governments might be willing to invest public money into the development of new vaccines and help prioritize public health initiatives to eliminate or reduce the burden of communicable diseases. Additionally, recent global public health events, such as COVID-19, have focused the world’s attention on how we evaluate infectious diseases. Economic assessments of vaccines have traditionally focused on applying incident cases to estimate direct and indirect costs in relation to outcome changes. What has lacked in these prior analyses, and what is most evident with COVID-19, has been the behavioral response and policy response which has caused tremendous economic consequences for households and governments. Recent estimates from the United States suggest local counties will lose $144 billion in revenue through 2021, with an additional expenditure of $30 billion attributed to COVID-19 [35]. This would suggest that in the event that a coronavirus vaccine costs $1 billion to develop, the fiscal ROI from preventing or reducing the effects of COVID-19 could be >100. The learnings from developing a vaccine for COVID-19 could also have spillover effects that could inspire development of subsequent vaccines to protect against existing coronaviruses in circulation and the emergence of future pathogens.

There are several limitations that need to be taken into consideration when developing and implementing fiscal models. Firstly, the modeling framework, like other government cost-benefit analyses, is based on assumptions regarding future economic conditions where changes in productivity, economic growth, and wages can influence future forecasts. Due to uncertainty about the future, these assumptions are held constant in the analysis, although it is recognized that changes in macroeconomic parameters in the future could seriously influence the conclusions drawn here. Secondly, fiscal models require some understanding of how transactions between citizens and their government transpire over the life-course. This data can be identified in most countries with varying quality. However, many low- and middle-income countries have limited or almost non-existent tax collection systems which precludes the opportunity for assessing fiscal value for governments. Furthermore, the output from vaccine fiscal models can inform government of the yield attributed to different national immunization programs. However, fiscal models only reflect transactions between citizens and state that are linked to productivity changes, failing to reflect societal preferences for different health conditions, consequently undervaluing the full value of immunization that may be captured in cost–utility analysis. In this respect, fiscal modeling provides a comprehensive picture of how preventing future infections can influence governments, however it does not fully capture the health benefits using conventional patient preference measures and impact on households. In addition, the fiscal models described here are developed based on single cohorts of vaccinated individuals. Taking into consideration the role that vaccines can have on economic growth [30], this would suggest that any expansion in the macroeconomy and associated tax revenue gains attributable to vaccination would not be captured within a cohort model, suggesting these values may be underestimates.

Fiscal analysis of vaccination programs has been conducted across a wide range of countries including both developed and developing economies [1,25,29,36]. The essential components for constructing fiscal models are national tax systems collecting direct and indirect taxes and some understanding of transfer payments made by government to citizens. Tax systems do vary across countries; however, most rely on the taxation of wages at source, and collection of value added taxes at the point of consumption which can be determined from government documents and intergovernmental organizations such as the Organisation for Economic Cooperation and Development (OECD). As tax evasion is prevalent in most countries, it is also possible to adjust tax revenue accordingly [24]. In terms of public benefits which vary between countries, there is often data available in all countries on the types of benefits that governments provide for maintaining living standards such as allowances, disability, and pensions [13]. Details of the model inputs and likely data sources for conducting fiscal analysis of vaccines are described in the Good Practices guideline published by the International Society for Pharmacoeconomics and Outcomes Research [13].

## 7. Conclusions

The fiscal modeling framework offers a valuable contribution to help consider the broader cross-sectorial impact of vaccines on government public accounts. Whilst this review has focused on vaccines, the framework is not exclusive to evaluating vaccines and can be used to assess a range of health programs funded by government. We propose that the fiscal modeling framework can be used alongside other health economic evaluations that can help to inform a range of stakeholders, especially those responsible for funding future and ongoing programs. Additionally, following the ambition of generational accounting that all public health programs need to be paid for by current and/or future generations, we believe this approach can help to address intergenerational fairness and partially address health system sustainability.

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
