# Peer review of "Estimating the Fiscal Consequences of National Immunization Programs Using a “Government Perspective” Public Economic Framework"

_vaccines, 2020, doi:10.3390/vaccines8030495_

Round 1

Reviewer 1 Report

In this manuscript, the authors evaluate the fiscal effects of national immunization programs with the goal of using these data to inform governmental decisions on vaccine resource distribution. They make a strong statement to support  the contention that governments should  at least partially prioritize vaccine allocation decisions to maximize fiscal returns and clearly delineate analytic terminology such as return on investment (ROI) as factors that should be considered in making vaccine decisions.  The authors use two specific vaccines as examples to illustrate the importance of their value-based evaluations, rotavirus vaccination in children and human papillomavirus vaccination in adolescents.  Finally, adult immunization programs for infectious diseases such as seasonal influenza, diphtheria, tetanus and herpes zoster are  demonstrated to yield at least a four-fold benefit for governments.

The manuscript is considered well written and informative. There are additional topics that could be discussed, which might contribute to our understanding of the fiscal consequences of vaccine decisions.  First, perhaps the authors should discuss the fiscal consequences of diseases for which an effective vaccine is lacking.  Certainly, in this regard, HIV AIDS and respiratory syncytial virus come to mind and I’m sure there are others.  What is the cost to governments of the lack of vaccines for these diseases?  This might serve to illustrate the points being made here.  Also, while the ultimate cost of the Covid-19 pandemic is far from being set, it might behoove the authors to raise this topic in the discussion.  They could use the tremendous economic effects of the ongoing pandemic worldwide to illustrate the extreme costs of the lack of a vaccine for this virus and, in turn, the savings that would derive from the development of an efficacious one.  After all, this is the most devastating pandemic in one hundred years. The authors should add it to the discussion, even though final data is not yet available.  The point can easily be made that for every dollar spent on a SARS-CoV-2 vaccine the return would be several fold. One can even reasonably make the point that the successful vaccine strategies developed for this virus could accelerate vaccine development for any future coronaviruses that might emerge, thus adding to their ROI. This is especially true since, with SARS, MERS and SARS-CoV-2, there  have been three pathogenic coronaviruses that have emerged just within the past two decades.  It is felt that the addition of these topics even in general terms would strengthen the manuscript, even though there may not be hard data available for them yet.

Author Response

Reviewer 1

In this manuscript, the authors evaluate the fiscal effects of national immunization programs with the goal of using these data to inform governmental decisions on vaccine resource distribution. They make a strong statement to support  the contention that governments should  at least partially prioritize vaccine allocation decisions to maximize fiscal returns and clearly delineate analytic terminology such as return on investment (ROI) as factors that should be considered in making vaccine decisions.  The authors use two specific vaccines as examples to illustrate the importance of their value-based evaluations, rotavirus vaccination in children and human papillomavirus vaccination in adolescents.  Finally, adult immunization programs for infectious diseases such as seasonal influenza, diphtheria, tetanus and herpes zoster are  demonstrated to yield at least a four-fold benefit for governments.

The manuscript is considered well written and informative. There are additional topics that could be discussed, which might contribute to our understanding of the fiscal consequences of vaccine decisions.  First, perhaps the authors should discuss the fiscal consequences of diseases for which an effective vaccine is lacking.  Certainly, in this regard, HIV AIDS and respiratory syncytial virus come to mind and I’m sure there are others.  What is the cost to governments of the lack of vaccines for these diseases?  This might serve to illustrate the points being made here. 

This is a valid point. We have added a statement regarding communicable disease for which no vaccines are currently available. You are correct in asserting that data is not available, but it is an important concept to note in our paper.

Also, while the ultimate cost of the Covid-19 pandemic is far from being set, it might behoove the authors to raise this topic in the discussion.  They could use the tremendous economic effects of the ongoing pandemic worldwide to illustrate the extreme costs of the lack of a vaccine for this virus and, in turn, the savings that would derive from the development of an efficacious one.  After all, this is the most devastating pandemic in one hundred years. The authors should add it to the discussion, even though final data is not yet available. 

We have addressed this point in the Discussion. We have also provided a reference that describes current fiscal projections in lost tax for the US.

The point can easily be made that for every dollar spent on a SARS-CoV-2 vaccine the return would be several fold. One can even reasonably make the point that the successful vaccine strategies developed for this virus could accelerate vaccine development for any future coronaviruses that might emerge, thus adding to their ROI. This is especially true since, with SARS, MERS and SARS-CoV-2, there  have been three pathogenic coronaviruses that have emerged just within the past two decades.  It is felt that the addition of these topics even in general terms would strengthen the manuscript, even though there may not be hard data available for them yet.

This is a good point. We have added this to the Discussion. Furthermore, we have put into context the vaccine development costs and potential ROI for COVID-19.

Reviewer 2 Report

This paper lays out a “fiscal model framework” that can be used to evaluate the cost-benefit ratio of a particular country’s vaccination strategy. The paper offers some important concepts, but I feel that some key changes would make it more accessible to the journal’s audience.

First, there are some styling inconsistencies throughout the paper. Here are some examples:

  • ”Modeling” and “modelling” are inconsistently used throughout the paper. One spelling should be used.
  • Lines 123-125: There are minor capitalization inconsistencies, and “i.e.” is not used consistently. As an example, “Incomings - Outgoings” is capitalized but shouldn’t be. Additionally, at the end of line 123, the clarification, “i.e. fiscal cash flow”, is not surrounded by parenthesis, whereas other i.e. clarifications in this same paragraph are. Additionally, “i.e.” should almost always be followed by a comma (e.g., “i.e., fiscal cash flow”).
  • There are stray commas in some locations. For example, see lines 51 and 57. There are other examples.

I suggest that the authors have a 3rd party read the paper with a keen eye towards minor grammar and punctuation inconsistencies.

More substantive issues:

  • Line 14 states that the authors received no funding to write this paper. Is this actually true? The authors wrote this paper in their spare time using no grant money?
  • Lines 116-134 are key to the paper, as they lay out the fiscal modeling framework. However, the equation on line 129 is not clearly described. Lines 120-128 and 130-134 describe some components of the equation, but I have a tough time truly understanding the variables in the equation. As an example, r is the “discount rate”, but it’s not clear from the text what this actually means or how it’s computed. The equation uses (1+r)t, and I don’t understand why 1 is added to r, and I don’t follow why it’s exponentiated by t. I think a more pointed description of this equation and each of its variables and components would be useful.
  • Related to the bullet above, there’s no discussion on data and how to actually use this model in practice. To evaluate this model, data must be gathered and used, and it’s not clear how difficult or easy this is in practice. As an example:
    • Presumably, each country and each disease/vaccine studied will require their own datasets. If I wanted to, for example, assign a value to Rt for rotavirus in Egypt, what specific data streams would be required?
    • When data streams aren’t available to directly evaluate the model, I imagine that proxy data streams are used to generate approximations. Is there an example of this?
    • How granular are the data? For example, what if I wanted to evaluate this model each week for each county in the U.S.; could I do that? Or would I be restricted to country-year-level data?
  • It’s not actually clear that the described fiscal model framework is actually used throughout the remainder of the paper. The 3 demonstrated use cases (rotavirus, HPV, and adult immunization) don’t specifically discuss how the fiscal model framework was used to arrive at the provided conclusions/analyses. A walkthrough of how the fiscal model framework is used in each of these 3 applications would help to drive home how the model can be used in practice (e.g., which data streams are useful, difficulties in evaluating the model in a given context).
  • In the rotavirus application description (line 164), the authors refer to a “social accounting matrix” without defining it. This seems like a separate tool from the proposed fiscal model framework for evaluating fiscal consequences. How is this related to the proposed model in the paper?
  • Lines 148-162: Throughout this paragraph, computed numbers are provided without any interpretation. As an example, I understand that Egypt’s breakeven age is 22 and Vietnam’s is 43. In practice, what does this difference actually mean? How should I interpret that? As another example, we’re told that Egypt’s fBCR increases over time to 1.2 from 0 and eventually becomes 2.8 at age 50; what does this actually mean?
  • Is there a particular reason why the 3 application areas were selected? I understand that they demonstrate uses of the proposed fiscal modeling framework, but are there other applications that were left out? Or are these the only applications that have been demonstrated so far?
  • How are the R&D costs of a vaccine factored into the model? I’ve read that an average vaccine that ends up being used in practice can cost somewhere in the neighborhood of $500 million (from basic research to testing and evaluation to scaled-up production). I’m not sure if this is relevant for the proposed fiscal model framework, but those costs aren’t insignificant, and an interesting component of R&D costs is that one country might front the costs of R&D, while another just benefits from that country’s work without putting in its own R&D efforts.
  • Lines 239-240: The authors state that “Nothing is lost by employing fiscal models for evaluating vaccines...”, but is this actually true? Are there really no downsides at all to this model?

Author Response

Reviewer 2

This paper lays out a “fiscal model framework” that can be used to evaluate the cost-benefit ratio of a particular country’s vaccination strategy. The paper offers some important concepts, but I feel that some key changes would make it more accessible to the journal’s audience.

First, there are some styling inconsistencies throughout the paper. Here are some examples:

”Modeling” and “modelling” are inconsistently used throughout the paper. One spelling should be used.

This has been corrected.

Lines 123-125: There are minor capitalization inconsistencies, and “i.e.” is not used consistently. As an example, “Incomings - Outgoings” is capitalized but shouldn’t be. Additionally, at the end of line 123, the clarification, “i.e. fiscal cash flow”, is not surrounded by parenthesis, whereas other i.e. clarifications in this same paragraph are. Additionally, “i.e.” should almost always be followed by a comma (e.g., “i.e., fiscal cash flow”).

This has been corrected.

There are stray commas in some locations. For example, see lines 51 and 57. There are other examples.

I suggest that the authors have a 3rd party read the paper with a keen eye towards minor grammar and punctuation inconsistencies.

This has been corrected and reviewed.

More substantive issues:

Line 14 states that the authors received no funding to write this paper. Is this actually true? The authors wrote this paper in their spare time using no grant money?

Correct no funding was received for preparing this manuscript. There are no promotional materials described in this paper. This paper reflects more than 20 years of combined experience of the authors conducting public economic assessments in health.

Lines 116-134 are key to the paper, as they lay out the fiscal modeling framework. However, the equation on line 129 is not clearly described. Lines 120-128 and 130-134 describe some components of the equation, but I have a tough time truly understanding the variables in the equation. As an example, r is the “discount rate”, but it’s not clear from the text what this actually means or how it’s computed. The equation uses (1+r)t, and I don’t understand why 1 is added to r, and I don’t follow why it’s exponentiated by t. I think a more pointed description of this equation and each of its variables and components would be useful.

We have further expanded this section in the revised manuscript.

Related to the bullet above, there’s no discussion on data and how to actually use this model in practice. To evaluate this model, data must be gathered and used, and it’s not clear how difficult or easy this is in practice. As an example:

This paper is written as a concept paper where fiscal modeling examples are described based on previous publications. The citations from the original sources are provided for interested readers to explore in more detail the model inputs. Furthermore, interested readers are able to access the Good Practices guideline for fiscal modeling in which a reference is provided.

Presumably, each country and each disease/vaccine studied will require their own datasets. If I wanted to, for example, assign a value to Rt for rotavirus in Egypt, what specific data streams would be required?

Yes, each fiscal model is reliant on country level data used to populate the model. This is consistent with other health economic modeling approaches requiring local cost data and population level data inputs. We have added text describing data requirements and potential data search strategies.

When data streams aren’t available to directly evaluate the model, I imagine that proxy data streams are used to generate approximations. Is there an example of this?

Fiscal analysis is performed using the trace from cost-effectiveness models for different vaccine preventable conditions. So essentially you need the disease modeling component from which the fiscal effects are determined.

We have described available data sources in the revised manuscript and provide appropriate references.

How granular are the data? For example, what if I wanted to evaluate this model each week for each county in the U.S.; could I do that? Or would I be restricted to country-year-level data?

Technically speaking a fiscal analysis can be performed at any geographical level provided that you have data to perform such an analysis. For example, you could perform an analysis at the level of city, county, region, state or nation. This all depends on the data that you have available. You would need specific data on age-specific taxes and transfer payments to answer the relevant policy question defined by the geography.

It’s not actually clear that the described fiscal model framework is actually used throughout the remainder of the paper.

The Discussion section has focused on fiscal analysis and its application. We do provide some theory to fiscal analysis and we also compare the approach with cost-effectiveness analysis which is likely to be a more widely known methodology. We also discuss how one might employ fiscal models in decision-making. So, in fact the entire Discussion is addressing fiscal analysis.

The 3 demonstrated use cases (rotavirus, HPV, and adult immunization) don’t specifically discuss how the fiscal model framework was used to arrive at the provided conclusions/analyses. A walkthrough of how the fiscal model framework is used in each of these 3 applications would help to drive home how the model can be used in practice (e.g., which data streams are useful, difficulties in evaluating the model in a given context).

The focus of this manuscript is to illustrate the application of fiscal modeling applied to vaccination programs. We provide some examples with rotavirus and HPV; however the aim of this paper is not to advocate for rotavirus and HPV. Hence, our concluding paragraph focuses on some of the broader issues to consider in relation to fiscal analysis.  We have updated the text to briefly describe the methods used in the cited papers.

In the rotavirus application description (line 164), the authors refer to a “social accounting matrix” without defining it. This seems like a separate tool from the proposed fiscal model framework for evaluating fiscal consequences. How is this related to the proposed model in the paper?

We have provided additional details regarding the social accounting matrix format.

Lines 148-162: Throughout this paragraph, computed numbers are provided without any interpretation. As an example, I understand that Egypt’s breakeven age is 22 and Vietnam’s is 43. In practice, what does this difference actually mean? How should I interpret that? As another example, we’re told that Egypt’s fBCR increases over time to 1.2 from 0 and eventually becomes 2.8 at age 50; what does this actually mean?

Fiscal models are versatile and can generate a range of outcomes. We have added a paragraph to the Discussion section to describe what we believe is the most important output; the fBCR. We also highlight that these figures can’t be transferable to other countries.

Is there a particular reason why the 3 application areas were selected? I understand that they demonstrate uses of the proposed fiscal modeling framework, but are there other applications that were left out? Or are these the only applications that have been demonstrated so far?

We provided examples for developed and developing economies to illustrate the versatility of the method. This has been added to the Discussion section.

How are the R&D costs of a vaccine factored into the model? I’ve read that an average vaccine that ends up being used in practice can cost somewhere in the neighborhood of $500 million (from basic research to testing and evaluation to scaled-up production). I’m not sure if this is relevant for the proposed fiscal model framework, but those costs aren’t insignificant, and an interesting component of R&D costs is that one country might front the costs of R&D, while another just benefits from that country’s work without putting in its own R&D efforts.

We have provided a worked example of how this could be done using fiscal models in the Discussion. It is very top-level approach to help illustrate.

Lines 239-240: The authors state that “Nothing is lost by employing fiscal models for evaluating vaccines...”, but is this actually true? Are there really no downsides at all to this model?

Generally, decision-makers value more information over less. The fiscal approach adds one additional piece of information to inform resource allocation [1]. We don’t advocate that this approach should replace cost-effectiveness analysis. It provides information for other stakeholders, particularly those responsible for funding vaccines with public money.

[1] J. Mauskopf, B. Standaert, M.P. Connolly, A.J. Culyer, L.P. Garrison, R. Hutubessy, M. Jit, R. Pitman, P. Revill, J.L. Severens, Economic analysis of vaccination programs, Value in health 21(10) (2018) 1133-1149.

Reviewer 3 Report

Dear Authors, 

The manuscript titled "Estimating the fiscal consequences of national immunization programs using a “government perspective” public economic framework" has been well written and logically constructed.

The concept you clearly state is interesting but should be better focused on different countries or continents.

The manuscript should be better clarify the differences of vaccination fiscal system in countries or continents since there are differences in economics, politics and government. Many countries have high taxes and vaccinations are free. On the other hand, some countries have low taxes and so the healthcare costs (including vaccinations) are very high.

it should be increased with references the "Adult vaccination" section. Indeed, some countries are linked to adult work and vaccinations could increase the economic growth as reported in a paper previously published (Quilici et al., 2015: Role of vaccination in economic growth. doi:10.3402/jmahp.v3.27044).

Author Response

Reviewer 3

The manuscript titled "Estimating the fiscal consequences of national immunization programs using a “government perspective” public economic framework" has been well written and logically constructed.

The concept you clearly state is interesting but should be better focused on different countries or continents.

We have provided some description regarding the application of the method to other markets.

The manuscript should be better clarify the differences of vaccination fiscal system in countries or continents since there are differences in economics, politics and government. Many countries have high taxes and vaccinations are free. On the other hand, some countries have low taxes and so the healthcare costs (including vaccinations) are very high.

A paragraph has been added to the Discussion to make this important point.

it should be increased with references the "Adult vaccination" section. Indeed, some countries are linked to adult work and vaccinations could increase the economic growth as reported in a paper previously published (Quilici et al., 2015: Role of vaccination in economic growth. doi:10.3402/jmahp.v3.27044).

We have added the main point of the paper by Quilici regarding economic growth to our manuscript.

Round 2

Reviewer 1 Report

In this manuscript, the authors evaluate the fiscal effects of national immunization programs with the goal of using these data to inform governmental decisions on vaccine resource distribution. They make a very strong case for the cost-effectiveness of vaccine development, using the examples of rotavirus vaccination in children and human papillomavirus vaccination in adolescents to make their point.  Also, adult immunization programs for infectious diseases such as seasonal influenza, diphtheria, tetanus and herpes zoster are  demonstrated to yield at least a four-fold benefit for governments. In this revision, the authors have added additional information suggested by one of the reviewers.  To this end, they address the enormous financial benefit to be derived from the development of a vaccine for the Covid-19 virus that is presently overwhelming the world’s population, suggesting that the ROI for such a vaccine could be >100.  These additions increase the scope of the review and it is considered appropriate for publication.

Author Response

We don't see any additional comments in relation to the resubmitted manuscript.

thanks for your comments. 

Reviewer 2 Report

I appreciate the authors' quick and thorough response to reviewers' comments. I feel that the paper has definitely improved, but I have just a few remaining comments:

  • I still feel like there's minimal discussion of how to actually interpret the use the equation on line 134. I understand the numerator in the summation, but I don't understand the denominator or its components, and I don't understand how the numerator relates to the denominator.
    • The "discount rate" is still not defined anywhere in the text. At this point, I suspect that this is economics jargon, but neither I nor most of the readers of this journal (I suspect) will know what this is. Defining all economics jargon would really help readability.
    • In the denominator, why is 1 added to the discount rate? And then why is (1+r) exponentiated by t?
      • When the variables are briefly defined below the equation, it's mentioned that (1+r)-t is the "discount formula", but it's not clear what this means or how it relates to the numerator or how it was derived.
    • If there were a dedicated paragraph for describing the equation more thoroughly in simple English without jargon, readability would be dramatically improved; merely defining the variables isn't enough. Rather, if the paper walked through the different components of the equation and described what they intuitively represent, the reader would be able to walk away with a clear understanding of the equation.
  • I still feel that a single example of how this equation should be used in practice is warranted. Even a toy example would help. I realize that the authors cite other papers that do this more in depth, but for this paper to truly stand on its own, a discussion and example of data and how to actually use the equation would really help an uninformed reader (such as myself).
  • This is a more minor issue, but there are still some punctuation and capitalization issues throughout the paper. I suspect that the journal editor will help address these, but a 3rd party that hasn't yet read the paper may also be able to help.

Author Response

Reviewer 2

Round 2 comments

I appreciate the authors' quick and thorough response to reviewers' comments. I feel that the paper has definitely improved, but I have just a few remaining comments:

It appears that most of the questions are in relation to the concept of discounting. This is an important element as it influences the results over the duration of the analysis. We have provided some additional background and explanation to help readers grasp this concept.

I still feel like there's minimal discussion of how to actually interpret the use the equation on line 134. I understand the numerator in the summation, but I don't understand the denominator or its components, and I don't understand how the numerator relates to the denominator.

Additional details have been added.

The "discount rate" is still not defined anywhere in the text. At this point, I suspect that this is economics jargon, but neither I nor most of the readers of this journal (I suspect) will know what this is. Defining all economics jargon would really help readability.

We have added 3 additional sentences to help explain the purpose of discounting and the potential range to apply. This is supported by 2 additional references.

In the denominator, why is 1 added to the discount rate? And then why is (1+r) exponentiated by t?

When the variables are briefly defined below the equation, it's mentioned that (1+r)-t is the "discount formula", but it's not clear what this means or how it relates to the numerator or how it was derived.

In our analysis we describe the standard formula for discounting. The discount rate, often denoted r, is used to generate a discounting factor D in the following way: D = [1/(1 + r)]^t. Using this formula, a discounting factor for any point in the future is obtained which can then be applied to benefits that occur at that point in the future[1]. Discounting is a well-established concept in all areas of finance and health economics so we don’t believe it is necessary to provide a detailed description beyond what is already provided.

We have provided a description of the concept of discounting in the revised manuscript.

If there were a dedicated paragraph for describing the equation more thoroughly in simple English without jargon, readability would be dramatically improved; merely defining the variables isn't enough. Rather, if the paper walked through the different components of the equation and described what they intuitively represent, the reader would be able to walk away with a clear understanding of the equation.

We have provided additional description of the variables used in the analysis and the origin of these inputs.

I still feel that a single example of how this equation should be used in practice is warranted. Even a toy example would help. I realize that the authors cite other papers that do this more in depth, but for this paper to truly stand on its own, a discussion and example of data and how to actually use the equation would really help an uninformed reader (such as myself).

We have provided a simplified example to illustrate the application of the fiscal modeling framework. We are not sure if this adds to the overall manuscript, but it can be considered for inclusion by the Editors.

This is a more minor issue, but there are still some punctuation and capitalization issues throughout the paper. I suspect that the journal editor will help address these, but a 3rd party that hasn't yet read the paper may also be able to help.

Thanks for noting this. Our colleague has reviewed the document and provided additional editorial feedback.

[1] Conrad Heilmann. Values in Time Discounting. Sci Eng Ethics. 2017; 23(5): 1333–1349. https://www.ncbi.nlm.nih.gov/pmc/articles/PMC5636865/

Round 3

Reviewer 2 Report

I really appreciate the authors' quick response to my comments. I especially appreciate the additional text above the equation to help explain each of the components. The added toy example paragraph helps me have a much more intuitive understanding of the equation. Thank you very much.

The only aspect that I must emphatically disagree with the authors on is this:

"Discounting is a well-established concept in all areas of finance and health economics so we don’t believe it is necessary to provide a detailed description beyond what is already provided."

As you can see at https://www.mdpi.com/journal/vaccines/about, Vaccines is not geared towards finance and health economics; it's generally geared more towards the biological aspects of vaccine development and use. As a result, the audience is very unlikely to have the same sort of understanding of finance and economics as the authors of this article. All economics and finance jargon must be clearly defined for the audience of this journal to avoid any confusion. I wouldn't submit a paper to an economics journal that involved non-trivial biological concepts without defining those concepts clearly for that audience.

Author Response

The revised manuscript describes the theory of discounting and we provide information regarding why it is necessary to perform discounting in economic analysis.

Additionally, we provided the range of discount rates that are often used and provided a citation. 

Furthermore, we have described the equation and how it is applied in the analysis. 

This has all been added to the Methods section. 

Thanks!

Mark